# A Quantitative Analysis Study on the Effects of Moisture and Light Source on FTIR Fingerprint Image Quality

**DOI:** 10.3390/s25041276

**Published:** 2025-02-19

**Authors:** Manjae Shin, Seungbong Lee, Seungbin Baek, Sunghoon Lee, Sungmin Kim

**Affiliations:** 1Department of Biomedical Engineering, Dongguk University Biomedical Campus, 32, Dongguk-ro, Ilsandong-gu, Goyang-si 10326, Republic of Korea; tlsakswo97@dongguk.edu (M.S.); sbleem9@gmail.com (S.L.); 2Department of Medical Device Business, Dongguk University, 30, Pildong-ro 1-gill, Jung-gu, Seoul 04620, Republic of Korea; bjh5863@gmail.com (S.B.); lsh1202lsh@dongguk.edu (S.L.)

**Keywords:** biometric, fingerprint, frustrated total internal reflection (FTIR), clinical study, statistical analysis

## Abstract

The frustrated total internal reflection (FTIR) optical fingerprint scanning method is widely used due to its cost-effectiveness. However, fingerprint image quality is highly dependent on fingertip surface conditions, with moisture generally considered a degrading factor. Interestingly, a prior study reported that excessive moisture may improve image quality, though their findings were based on qualitative observations, necessitating further quantitative analysis. Additionally, since the FTIR method relies on optical principles, image quality is also influenced by the wavelength of the light source. In this study, we conducted a preliminary clinical experiment to quantitatively analyze the impact of moisture levels on fingertips (wet, dry, and control) and light wavelengths (red, green, and blue) on FTIR fingerprint image quality. A total of 20 male and female participants with no physical impairments were involved. The results suggest that FTIR fingerprint image quality may improve under wet conditions and when illuminated with green and blue light sources compared to dry conditions and red light. Statistical evidence supports this consistent trend. However, given the limited sample size, the statistical validity and generalizability of these findings should be interpreted with caution. These insights provide a basis for optimizing fingerprint imaging conditions, potentially enhancing the reliability and accuracy of automatic fingerprint identification systems (AFIS) by reducing variations in individual fingerprint quality.

## 1. Introduction

Fingerprints are swirly and complex patterns formed by the protrusion of sweat glands inside the fingertip tissue, developing around seventh month of fetal growth. These patterns are innate from birth and remain unique to each individual, even identical twins who share many of their genes, and they remain unchanged throughout life unless they are damaged by external factors such as wounds and scars. These properties and the relative ease of acquisition compared to other biometric traits make it one of the most widely used in the field of biometric authentication worldwide [1,2,3,4,5,6].

Acquiring fingerprints from subjects is a crucial step in the identification process. Traditionally, an analog method known as the ink technique was commonly used, in which the hand was inked and pressed onto a paper card, a process classified as off-line sensing. With advancements in digital information processing and computing technology, live-scan methods, which capture digital fingerprint data, have largely replaced the existing approach in most applications. These modern methods eliminate the need for direct ink contact and vary widely depending on the underlying sensor technology, typically categorized into optical, solid-state, and ultrasonic sensors [3,7].

Optical and solid-state fingerprint sensors, such as frustrated total internal reflection (FTIR) and capacitive sensors, have been noted for their image quality being significantly affected by the condition of the skin surface. In contrast, ultrasonic fingerprint sensors leverage the properties of ultrasound waves, which can propagate through tissue and reach the dermal layer. This capability allows ultrasonic sensors to capture not only the surface features of the finger but also the dermal papillae, which corresponds to the actual fingerprint pattern. As a result, ultrasonic fingerprint sensors offer greater robustness against the presence of moisture and debris on the fingertip, enhancing reliability in less-than-ideal conditions. However, ultrasonic fingerprint sensors also have some downsides: they are relatively expensive compared to most optical and solid-state scanners and exhibit slower scanning speed due to the need to transmit pulses and wait for the return signal [3,6,8,9].

As previously discussed, optical fingerprint sensors have issues related to image quality caused by moisture, debris, and defects on the skin and sensor surface. Nevertheless, optical fingerprint systems are still widely used today as cutting-edge image sensors equipped with high resolution are available on the market at a relatively low price due to advances in digital image sensor technology [3,6,10]. According to a prior study by Smith et al. [11], excessive moisture on the fingerprint surface, previously considered a factor that degrades the image quality of optical FTIR fingerprint imaging methods, was found to improve fingerprint image quality, contrary to prior understanding. However, this finding has some limitations, as it relies on qualitative observations conducted by the researchers without incorporating a qualitative approach to enhance the reliability of the results.

In this paper, a preliminary clinical study was conducted to statistically analyze the quality of FTIR fingerprint images at varying moisture levels on the fingerprint surfaces, utilizing fingerprint feature points for quantitative assessment. Additionally, we investigated the effects of light wavelength (color), considering that inherent characteristic of the optical system, which affects fingerprint image quality by influencing the interaction between light and the optical properties of human skin [3,12,13].

## 2. Materials and Methods

In this study, fingerprint images of participants were collected using a low-cost in-house FTIR fingerprint imaging device. This apparatus utilizes a transparent quartz (SiO_2_) plate to induce the FTIR phenomenon, which arises due to direct contact between the users’ fingers and the quartz plate. RGB LEDs are positioned along the side of the quartz plate to serve as a light source. The device features a 3D printed structure and employs a CMOS color image sensor (HIKVISION, Hangzhou, China; MV-CB070-10UC-C camera, equipped with a Sony ExMor R CMOS image sensor 3072 × 2048) to capture fingerprint images. This apparatus is controlled via Arduino Nano and a PC to synchronize LED switching and exposure of image sensors, as shown in Figure 1. Acquired FTIR fingerprint images using this apparatus contain not only fingerprint patterns but also unnecessary background and noise caused by various factors, making a proper image processing procedure essential.

### 2.1. Valid Channel Data and Region of Interest (ROI) Extraction

Fingerprint recognition systems return various types of fingerprint data depending on the type of sensor and the principles employed. In the case of optical systems, grayscale or color images can be obtained depending on the image sensor used. The proposed study utilized a color image sensor, which allows for the acquisition of 3-channeled color images composed of red, green, and blue channels respectively. However, this multidimensional image format poses a challenge as it is not suitable for most image processing operations. To convert a 3-channel color image into a monochrome grayscale image, the average value of each channel is commonly used, as shown in Equation (1) [14](1)$$I_{grayscale}\left(i,j\right)=\frac{I_{R}\left(i,j\right)+I_{G}\left(i,j\right)+I_{B}\left(i,j\right)}{3}$$
where $I_{grayscale}\left(i,j\right)$ is converted grayscale image pixel, $I_{R}\left(i,j\right)$, $I_{G}\left(i,j\right)$ and $I_{B}\left(i,j\right)$ represent the red, green, and blue channel pixel data of the original color image respectively. However, according to Equation (1), since all three channels contribute to the resultant image, this averaging conversion method cannot independently reflect the effect of R, G, and B lighting conditions on the quality of the image. Therefore, in this paper, only the channel components corresponding to the main wavelength of the light source represented as R, G, and B colors were used, as demonstrated in Figure 2. After completing the channel data acquisition step, the region of interest (ROI) containing the fingerprint pattern was extracted. An ROI size of 1200 × 800 was utilized, based on the biggest fingerprint size of the specimen, which measured 37.5 mm × 25 mm.

### 2.2. Fingerprint Mask Generation

Extracted valid channel data from the previous step are composed of the foreground containing a fingerprint pattern and unnecessary background. FTIR fingerprint images are characterized by the fingerprint pattern areas in direct contact with the sensor surface appearing brightly illuminated, while the background remains relatively dark. Based on this characteristic, where the contact region typically forms an elliptical shape, the image binarization method can be applied to selectively generate a mask that separates the foreground of the FTIR fingerprint image. The valleys of the fingerprint pattern in the brightly illuminated foreground appear dark, which complicates the determination of an optimal threshold for generating an accurate mask. To address this issue, the fingerprint pattern component, characterized by its high-frequency nature, is suppressed using Gaussian blur, a widely recognized low-pass filtering method, as shown in Equations (2) and (3) [15].(2)$$G\left(u,v\right)=\frac{1}{2\pi \sigma^{2}}\;\cdot \;\exp\left(-\frac{u^{2}+v^{2}}{2\sigma^{2}}\right)$$(3)$$I^{\prime }\left(i,j\right)=\sum_{u=-\frac{\omega_{0}}{2}}^{\frac{\omega_{0}}{2}}\sum_{v=-\frac{\omega_{0}}{2}}^{\frac{\omega_{0}}{2}}I\left(i-u,i-v\right)\;\cdot \;G\left(u,v\right)$$G(u,v) represents a normalized Gaussian kernel of size $\omega_{0}\times \omega_{0}$, while $I\left(i,j\right)$ and $I^{\prime }\left(i,j\right)$ refer to the original image and the blurred image, respectively. In this study, the parameters σ and $\omega_{0}$ for the Gaussian kernel were empirically set to 50 and 99.

The Otsu thresholding technique is employed to automatically determine the binarization threshold for an image in which the fingerprint pattern component has been removed using Gaussian blur. Assuming the image consists of two classes—background and foreground—the algorithm analyzes the image histogram to identify an optimal threshold T that minimizes the intra-class variance of each of the two classes of background and foreground, which can be expressed as Equation (4) [16].(4)$$T^{*}={\mathrm{argmax}}_{T}\left[\omega_{1}\left(T\right)\omega_{2}\left(T\right){\left(\mu_{1}\left(T\right)-\mu_{2}\left(T\right)\right)}^{2}\right]$$In this context, $\omega_{1}\left(T\right)$ and $\omega_{2}\left(T\right)$ represents the probabilities of the foreground and background classes, respectively, for a given threshold T. Similarly, $\mu_{1}\left(T\right)$ and $\mu_{2}\left(T\right)$ denote the average values of each respective class. The threshold $T^{*}$ is defined as the value that satisfies Equation (4) among all possible thresholds T. The masks generated through the subsequent process are employed to facilitate efficient operations by selectively targeting pixels within the valid fingerprint pattern area, thereby enhancing the algorithm’s overall performance.

### 2.3. Local Brightness Error Correction

The image results obtained using the FTIR optical fingerprint imaging device are significantly influenced by the contact state and pressure between the fingerprint and the sensor. Unlike the sensor surface—represented here by a quartz plate, which has a two-dimensional flat shape—a fingerprint typically exhibits a three-dimensional structure with a central protrusion tapering toward the edges. When greater pressure is applied to the square quartz plate to capture a wider fingerprint area, the disparity in contact pressure between the central region and the protruding edges increases. This uneven pressure distribution leads to local brightness errors in the resulting image. Additionally, the complex interactions between the light source and the internal tissue of the finger or the blood result in certain areas of the fingerprint appearing relatively dark, as in Figure 3.

Local brightness errors in the fingerprint pattern area can significantly degrade the performance of image processing techniques that rely on pixel intensity values, necessitating appropriate correction. The masked fingerprint pattern image can be modeled as comprising two components: a ridge pattern component, which is the primary target for enhancement, and a background component, which corresponds to the imaged surface of the fingertip excluding the ridge pattern. This relationship is expressed in Equation (5).(5)$$I_{Masked}\left(i,j\right)=r\left(i,j\right)+b_{ideal}\left(i,j\right)$$$I_{Masked}$ represents a masked fingerprint image, r denotes the ridge pattern component, $b_{ideal}$ represents the ideal background component without local brightness errors, and i and j correspond to the pixel coordinates of the image. However, due to the various factors discussed earlier, local brightness errors occur in the fingerprint background area. The realistic fingerprint background component $b_{real}$, which accounts for these errors, can be expressed using Equations (6) and (7).(6)$$I_{Masked}\left(i,j\right)=r\left(i,j\right)+b_{real}\left(i,j\right)$$(7)$$b_{real}\left(i,j\right)=b_{ideal}\left(i,j\right)+e\left(i,j\right)$$e represents the local brightness error component found in $b_{real}$. To suppress the high-frequency r component in $I_{Masked}$ and separate b, a Gaussian blur is applied to $I_{Masked}$, as expressed in Equation (8). $b_{real}$, comprising e and $b_{ideal}$ as shown in Equation (9), is then subtracted from $I_{Masked}$ to isolate r, free of local brightness errors. The corrected image demonstrated in Figure 4 was obtained by adding the average pixel value of $b_{real}$ to r, as shown in Equation (10).(8)$$b_{real}=\sum_{u=-\frac{\omega_{0}}{2}}^{\frac{\omega_{0}}{2}}M\left(i-u,j-v\right)\cdot G\left(u,v\right)$$(9)$$r\left(i,j\right)=I_{Masked}\left(i,j\right)-b_{real}\left(i,j\right)$$(10)$$I_{Corrected}\left(i,j\right)=r\left(i,j\right)+\frac{1}{M\times N}\sum_{i=0}^{M-1}\sum_{j=0}^{N-1}b_{real}\left(i,j\right)$$

### 2.4. Normalization

The previous correction process for local brightness errors encounters significant deviations across result images due to an overall decrease in brightness and contrast, influenced by the original image’s pixel values. To reduce these deviations in the acquired fingerprint data and improve the consistency of images provided to algorithms that enhance fingerprint patterns, a normalization process, as defined in Equation (11), is required. This process adjusts the average pixel intensity and variance of the image to fixed values [14].(11)$$I_{Norm}(i,j)=\left\{\begin{array}{c} m_{0}+\sqrt{\frac{v_{0}\times {\left(I_{Corrected}\left(i,j\right)-m\right)}^{2}}{v}}\;\;\;\left(I_{Corrected}\left(i,j\right)\geq m\right)\\ m_{0}-\sqrt{\frac{v_{0}\times {\left(I_{Corrected}\left(i,j\right)-m\right)}^{2}}{v}}\;\;\;\left(I_{Corrected}\left(i,j\right)<m\right) \end{array}\right.$$m and v represent the average pixel intensity and variance of the current image, respectively, while $m_{0}$ and $v_{0}$ denote the target average pixel intensity and variance to be achieved through the normalization process. In this study, both $m_{0}$ and $v_{0}$ were set to 100, and the normalized image is shown in Figure 5.

### 2.5. Ridge Orientation Map Estimation

The ridge orientation map provides information about the direction of the fingerprint ridge pattern and is a critical component in generating the Gabor kernel used to enhance the fingerprint ridge pattern. To determine the orientation of the fingerprint ridge pattern, the pixel intensity gradient along the x-axis and y-axis is computed utilizing the Sobel operator, as defined in Equations (12) through (15) [17].(12)$${\mathrm{Sobel}}_{x}=\left[\begin{array}{ccc} 1 & 0 & -1\\ 2 & 0 & -2\\ 1 & 0 & -1 \end{array}\right]$$(13)$${\mathrm{Sobel}}_{y}=\left[\begin{array}{ccc} 1 & 2 & 1\\ 0 & 0 & 0\\ -1 & -2 & -1 \end{array}\right]$$(14)$$Gradient_{x}=I_{Norm}\left(i,j\right){\;*\;\mathrm{Sobel}}_{x}$$(15)$$Gradient_{y}=I_{Norm}\left(i,j\right)\;*\;Sobel_{y}$$$G_{x}\left(i,j\right)$ and $G_{y}\left(i,j\right)$ represent the gradient images in the x-axis and y-axis directions, respectively, as illustrated in Figure 6. Using two gradient images that represent the rate of change of adjacent pixel intensity of different directions, the angle perpendicular to the ridge pattern can be calculated. However, due to the varying directions of ridge patterns across different regions of a fingerprint, it is not feasible to determine a single direction for the entire fingerprint image. To address this issue, the image is divided into square sub-images of W×W size. For each block, the direction of the most dominant ridge pattern is determined through the process described in Equation (16) through Equation (18) [14,18].(16)$$V_{x}\left(i,j\right)=\sum_{u=i-\frac{W}{2}}^{i+\frac{W}{2}}\sum_{v=j-\frac{W}{2}}^{j+\frac{W}{2}}2\cdot G_{x}\left(u,v\right)\cdot G_{y}\left(u,v\right)$$(17)$$V_{y}\left(i,j\right)=\sum_{u=i-\frac{W}{2}}^{i+\frac{W}{2}}\sum_{v=j-\frac{W}{2}}^{j+\frac{W}{2}}{\left(G_{x}\left(u,v\right)\right)}^{2}-{\left(G_{y}\left(u,v\right)\right)}^{2}$$(18)$$\theta \left(i,j\right)=\frac{1}{2}\arctan\frac{V_{x}\left(i,j\right)}{V_{y}\left(i,j\right)}$$θ represents the ridge orientation map, while $V_{x}$ and $V_{y}$ indicate the directional component along the calculated row and column axes, respectively. However, errors may arise when the calculated θ significantly deviates from the actual ridge orientation due to defects in the fingerprint-capturing process or the fingerprint itself. To address this potential issue, the characteristic that ridge orientations in adjacent areas are generally consistent is utilized. Gaussian blur is employed for the elements of θ to correct mispredicted values, smoothing them to align with surrounding elements. Applying Gaussian blur requires transforming θ into a continuous vector field, as detailed in Equations (19) and (20).(19)$$\phi_{x}\left(i,j\right)=\cos\left(2\cdot \theta \left(i,j\right)\right)$$(20)$$\phi_{y}\left(i,j\right)=\sin\left(2\cdot \theta \left(i,j\right)\right)$$$\phi_{x}$ and $\phi_{y}$ represent continuous vector fields in the *x*-axis and *y*-axis directions, respectively. After applying Gaussian blur to the continuous vector fields, as described in Equations (21) and (22), the corrected ridge orientation map O can be calculated using Equation (23) [14].(21)$$\phi_{x}^{\prime }\left(i,j\right)=\sum_{i=u-\frac{W_{\Phi }}{2}}^{\frac{W_{\Phi }}{2}}\;\;\sum_{j=v-\frac{W_{\Phi }}{2}}^{\frac{W_{\Phi }}{2}}G\left(u,v\right)\cdot \phi_{x}\left(i-u,j-v\right)$$(22)$$\phi_{y}^{\prime }\left(i,j\right)=\sum_{i=u-\frac{W_{\Phi }}{2}}^{\frac{W_{\Phi }}{2}}\;\;\sum_{j=v-\frac{W_{\Phi }}{2}}^{\frac{W_{\Phi }}{2}}G\left(u,v\right)\cdot \phi_{y}\left(i-u,j-v\right)$$(23)Oi,j=12 arctan ⁡ϕy′i,jϕx′i,j$\phi_{x}^{\prime }$ and $\phi_{y}^{\prime }$ represent continuous vector field to which a Gaussian kernel of $W_{\Phi }\times W_{\Phi }$ size has been applied. Figure 7 provides a visualized example of the ridge orientation map O calculated through the entire process.

The white lines in Figure 7 represent the visualization of the ridge orientation map O. These lines are drawn from the center of each W×W sized sub-images, where the direction angle perpendicular to the ridge is converted into the slope of the straight line. In this study, W was set to 40, resulting in the entire image being divided into a 30 × 20 grid array.

### 2.6. Ridge Spatial Frequency Estimation

To generate the Gabor kernel used to enhance fingerprint ridge patterns, additional information on the spatial frequency of the ridge pattern is required. This information is derived from the average distance between adjacent ridges in a sub-image, along with the ridge orientation map O obtained in the ridge orientation map estimation step described above in Section 2.5 [14,19]. Since the spacing between adjacent ridges within each divided sub-image is not always consistent, the average distance between the observed ridges in the region is calculated and used as the spatial frequency parameter for the Gabor kernel. To estimate the average spatial frequency, the positions of the ridges in the sub-image and the distance between them must first be determined. This is achieved by rotating the 30 × 20 sub-image array element according to the corresponding component of $O\left(i,j\right)$, aligning all ridge patterns in a uniform direction. The aligned ridges are then projected along the row direction, as illustrated in Figure 8.

The positions of the ridges were detected by identifying the extrema of the projected data, as illustrated in Figure 8. Since ridges appear as bright regions in the FTIR fingerprint image, the spatial frequency was calculated by locating the maxima and averaging the distance between adjacent maxima. After computing the rate of change of the projection data, as described in Equation (24), the points where the rate of change reversed from positive (+) to negative (−) were identified as maxima.(24)$$K\left(i\right)=P\left(i+1\right)-P\left(i\right)$$P(i) represents the one-dimensional projection data, and K(i) denotes the rate of change between adjacent values in the projection data. To minimize errors caused by noise in the image, which could result in the misidentification of faux ridges, maxima located below a specific distance threshold were treated as noise and excluded. The average distance between the remaining adjacent maxima was then used as the spatial frequency parameter for the Gabor kernel.

### 2.7. Gabor Kernel Fingerprint Ridge Pattern Enhancement

The Gabor kernel is a well-known band-pass filter in signal and image processing, designed to amplify specific frequency bands while attenuating others. By leveraging these characteristics, the Gabor kernel can effectively enhance fingerprint patterns by amplifying the signal corresponding to the ridge pattern within each divided sub-image while suppressing irrelevant components. The Gabor kernel can be mathematically defined as in Equation (25) [14,18,20,21].(25)$$Gabor\left(x,y;\lambda ,\theta ,\psi ,\sigma ,\gamma \right)=\exp\left(-\frac{x^{\prime 2}+\gamma^{2}y^{\prime 2}}{2\sigma^{2}}\right)\cdot \cos\left(2\pi \frac{x^{\prime }}{\lambda }+\psi \right)\;\left(x^{\prime }=x\cos\theta +y\sin\theta ,\;y^{\prime }=-x\sin\theta +y\cos\theta \right)$$Here, $x^{\prime }$ and $y^{\prime }$ represent the image coordinates obtained by rotating the original coordinates x and y of the Gabor kernel according to $O\left(i,j\right)$. λ denotes the spatial frequency calculated in the previous step 2.6, while ψ represents the phase offset of the sinusoidal element of the Gabor kernel. Parameters σ and γ control the standard deviation and the spatial aspect ratio of the Gabor kernel, respectively. In this study, ψ=0 and γ=1 were used, and the fingerprint ridge pattern was enhanced as shown in Figure 9, using σ=0.4 λ experimentally. The blank black regions in Figure 9 indicate areas where the spatial frequency estimation failed within the divided sub-image.

### 2.8. Quantitative Evaluation of Fingerprint Image Quality

Each fingerprint possesses a distinct ridge pattern, but certain common structures can be identified across human fingerprints in detail, making categorization possible. These structures, referred to as minutiae in the field of fingerprint recognition, represent key feature points [2,3,22]. The typical types of fingerprint minutiae are illustrated in Figure 10.

Before the development of computer-based information processing technologies, fingerprint identification was conducted using analog methods. These methods involved indexing the location and types of minutiae, as illustrated in Figure 10, and manually analyzing them one by one. With advancements in technology, systems were developed to automate the process of extracting fingerprint minutiae. The earliest fingerprint minutiae search algorithms were designed to simulate the manual procedures of early fingerprint analysts. Today, automated fingerprint identification systems (AFIS) that determine identity by comparing the similarity of fingerprint minutiae between two fingerprints remain an active area of research and development [3,23].

High-quality fingerprint images in general AFIS are those where the ridge patterns and feature points are distinctively visible, with well-defined contrast between the ridges and valleys [3,24]. The quality evaluation of fingerprint images has primarily been conducted in a qualitative manner, relying on visually observable factors. Consequently, the process is prone to significant issues, including the potential involvement of the inspector’s subjectivity in assessing image quality. Furthermore, with the increasing volume of accumulated fingerprint datasets driven by advancements in data storage and processing technologies, the traditional qualitative evaluation method—in which fingerprint quality evaluators visually inspect and assess all images—has become impractical. This has led to a growing demand for quantitative methods to evaluate the quality of fingerprint images [25,26].

The National Institute of Standards and Technology (NIST), a government research organization under the U.S. Department of Commerce, plays a crucial role in establishing reliable standards and guidelines across various fields. Notably, it contributes significantly to the development of standards and performance assessments in biometric technologies, including fingerprint recognition. NIST defines the quality of fingerprint images as a *predictor of matching algorithms’ performance*, emphasizing the improvement in AFIS performance achieved by increasing both the number and the prediction accuracy of minutiae points extracted by the searching algorithm when high-quality fingerprint images are used. In other words, the quality of a fingerprint image can be quantitatively assessed by analyzing the number of extracted minutiae points and their prediction accuracy when the image is input into a fingerprint recognition system [27,28,29,30].

Based on this definition, the quality of fingerprint images in this study was evaluated by extracting two types of fingerprint minutiae using the NIST Fingerprint Minutiae Viewer (FpMV) fingerprint feature point visualization software: (a) ridge endings and (b) ridge bifurcations, as shown in Figure 10. The extracted minutiae were categorized into two groups: true positive (TP), which correctly identified the location and type, and false positive (FP), which incorrectly identified either the location or type. Additionally, the FpMV software assigns a quality indicator to each extracted fingerprint minutiae, ranging from 0.01 to 0.99, where 0 indicates background and values closer to 1 represent higher-quality feature points [29,30].

## 3. Results

### 3.1. Acquisition of Fingerprint Data

This study was conducted following formal approval from Dongguk University Institutional Review Board (IRB) (Approval No. DUIRB2024-06-02) to quantitatively evaluate the quality of FTIR fingerprint imaged based on variations in the moisture state of the fingerprint surface and the color of the light source used for imaging. Clinical data were obtained by recruiting male and female participants aged between their 20s and 50s, with no physical defects related to their fingers. Fingerprint images were collected from the thumbs and index fingers of each participant.

Using the in-house FTIR fingerprint imaging device, fingerprint images were sequentially collected under three conditions: untreated condition (UC), which served as the control group without any experimental manipulation; dry condition (DC), simulating an extremely dry fingerprint; and wet condition (WC), representing an excessively wet scenario. First, images were captured under UC, where no pre-treatment was applied. Next, for DC, the fingertip surface and quartz plate were thoroughly cleaned using an alcohol swab, microfiber cloths, and an air blower to remove any oily residue or small debris before imaging. Finally, for WC, the fingertip and quartz plate were cleaned again following the same procedure as in DC, after which 100 ul of distilled water was loaded on the quartz plate by micropipette before the fingerprint image was captured. In the experimental procedure, the same finger under the same moisture condition was captured separately under red (R), green (G), and blue (B) lighting conditions. To accurately compare the effect of light wavelength (color) on fingerprint image quality, the position of the participant’s fingertip remained unchanged, and the light sources were switched sequentially at 1 s intervals to ensure rapid and consistent image acquisition. To increase the dataset size, both the thumb and index fingers of participants were imaged in this sequence. Through this approach, fingerprint images obtained under DC and WC were acquired without contaminants such as dust particles and sebum. All fingerprint acquisitions followed a predefined routine to ensure consistency.

### 3.2. Moisture Conditions

Since fingerprints are unique to each individual, there is no direct correlation between the absolute number of minutiae points observed in fingerprint images across different participants. Therefore, this study focused on analyzing the difference in the number of minutiae points caused by moisture variations in the same fingers of the same participant. The consistency of this trend across the 20 participants was statistically verified. Specifically, the number of TP minutiae points with a quality index of 0.9 or higher, as captured under different moisture conditions, was analyzed.

A repeated measures one-way analysis of variance (RM one-way ANOVA) test was employed as the appropriate statistical analysis method to assess whether there were statistically significant differences and trends among the comparison groups. For groups showing significant differences in the RM one-way ANOVA test, a post hoc analysis was conducted to quantify the extent of the numerical differences [31]. Since the UC was set as the control group, a Dunnett multiple comparison (MC) was conducted using GraphPad Prism 10 software with its own *p*-value notation style [32]. Figure 11 and Figure 12 present the results of the RM one-way ANOVA analysis for the index finger and thumb, respectively, while Table 1 and Table 2 provide the Dunnett MC PostHoc results corresponding to Figure 11 and Figure 12.

The analysis of TP minutiae points with a quality index of 0.9 or higher revealed statistically significant differences in the number of TP minutiae points across different moisture conditions in all experimental results, except for the thumb under the R light condition. When comparing the UC and WC, WC consistently showed a tendency to have more TP points than UC. For the index finger, there was no significant difference in the number of TP points between UC and DC. However, for the thumb, DC exhibited a lower average number of TP minutiae points than UC under the G and B lighting conditions.

Furthermore, an analysis was conducted on the survival rate of the TP minutiae points, defined as the number of TP points remaining as the quality index became stricter. Analysis of all fingerprint data obtained in this study revealed that all detected minutiae points had a quality index of 0.7 or higher. Therefore, the number of TP points with quality indices of 0.7, 0.8, and 0.9 or higher were all summed, assigning greater weight to higher-quality minutiae points. Figure 13 and Figure 14 present the RM one-way ANOVA analysis results for the experimental data calculated in this manner, while Table 3 and Table 4 provide the corresponding Dunnet MC test results.

The analysis reflecting the survival rate provided statistical evidence that, for the index finger, more TP minutiae points were observed under WC conditions compared to UC. For the thumb, the results aligned with the previous analysis showing a strong statistical association where WC conditions produce more TP minutiae points than UC, while DC conditions resulted in fewer TP points compared to UC. Notably, the previous analysis method found no significant correlation between different moisture conditions for the thumb under the R light conditions. However, the survival rate analysis revealed a significant correlation in the number of TP minutiae points among DC, UC, and WC moisture conditions for the thumb with R lighting.

### 3.3. Results: Lighting Conditions

To quantitatively analyze the effects of light source conditions on the quality of FTIR fingerprint images, the number of TP minutiae points with a quality index of 0.9 or higher and the number of TP minutiae points reflecting the survival rate were examined using the same analysis method applied to moisture condition experiments. An RM one-way ANOVA test was conducted to determine whether statistically significant differences existed between independent light conditions. Post hoc analysis was performed to evaluate the specific numerical differences between the conflicting conditions that exhibited significant results. Unlike the moisture condition experiment, there was no control group among the light source conditions, as all conditions were independent. Therefore, a Tukey MC test was employed as the appropriate post hoc method. Figure 15 and Figure 16 present the RM one-way ANOVA test results for the index finger and thumb, respectively, while Table 5 and Table 6 display the corresponding Tukey MC test results.

The analysis of TP minutiae points with a quality index of 0.9 or higher under different light source conditions revealed statistically significant differences between the R lighting and the B lighting condition for both index finger and thumb. A consistent trend was observed, showing that the B light condition produced a higher number of TP minutiae points compared to the R light condition. Additionally, for the index finger, significant differences were found between the R and G light source conditions across all moisture conditions, with the G light source condition yielding more TP points than the R light source condition. For the thumb, similar results were observed but only under the WC moisture condition, where the G lighting condition also produced more TP minutiae points than the R lighting condition. No significant differences were found between the G and B light conditions for either the index finger or the thumb.

Next, the analysis of the number of TP minutiae points reflecting the survival rate was performed using the same method as the analysis for moisture conditions. Figure 17 and Figure 18 represent the RM one-way ANOVA test results for the index finger and thumb, respectively, while Table 7 and Table 8 provide the corresponding Tukey MC post hoc results.

The survival rate analysis revealed a statistically significant correlation under the DC moisture condition, where the G light source condition produced a higher number of TP minutiae points reflecting the survival rate compared to the R light source condition. However, no statistical significance between the three different light sources was observed among the other moisture conditions. For the thumb, strong statistical evidence indicated that more TP points were observed under the B and G lighting conditions for all moisture conditions compared to R lighting conditions, consistent with the findings from the simple comparative analysis of TP points with a quality index of 0.9 or higher.

## 4. Discussion

The evaluation of FTIR fingerprint image quality, based on the analysis of TP minutiae points with a quality index of 0.9 or higher for different moisture conditions, revealed that the WC moisture condition, characterized by excessive moisture on the fingerprint surface, consistently enhanced the quality of fingerprint images compared to the UC control group. In contrast, the DC moisture condition showed the potential for similar or even reduced quality compared to UC.

In the same manner, our findings indicate that the B and G lighting conditions significantly enhance fingerprint image quality compared to R light sources. This result can be explained by the interaction between light wavelength and the optical properties of human skin. Shorter wavelengths, such as B and G light, exhibit lower penetration depth in biological tissues, enhancing surface contrast and minimizing subsurface scattering. In contrast, longer wavelengths, such as R lighting conditions, penetrate deeper into the skin, increasing subsurface diffusion and potentially reducing ridge visibility and these findings align with previous studies [33,34]. However, no significant quality differences were identified between the G and B lighting conditions. Future studies should incorporate a more detailed theoretical analysis of the physical mechanisms underlying the impact of different light wavelengths on FTIR fingerprint image quality. This could involve modeling the absorption, reflection, and scattering properties of different wavelengths to better understand their interactions with fingerprint ridges and valleys, particularly focusing on the specific wavelength characteristics of G and B light sources. Furthermore, in this study, the illumination intensity was not considered as a variable, as all fingerprint images were captured using the maximum brightness available from a 12 V LED system. This limitation suggests that future studies should investigate not only spectral characteristics but also the effects of illumination intensity on FTIR fingerprint image quality to gain a more comprehensive understanding of its influence.

The additional analysis of TP minutiae points reflecting the survival rate for moisture conditions confirmed a consistent trend: the quality of fingerprint images improved under WC, while DC resulted in reduced image quality, aligning with the previous analysis results. Similarly, the analysis of light source conditions demonstrated that FTIR fingerprint images captured under the R light source condition exhibited lower quality.

The analysis revealed that statistically significant quality differences were expected under certain experimental conditions, but no clear correlation was found. First, for the thumb images captured under the R lighting condition, it is believed that the impact of moisture conditions on the quality of FTIR fingerprint images was overwhelmed due to the overall degradation of image quality caused by the R lighting condition. Second, in the analysis of light source conditions, the thumb images captured under DC and UC moisture conditions showed no quality difference between the G and R light source conditions. This can be attributed to the negative effect on image quality caused by the DC moisture conditions, as well as the low consistency in data from the UC control group, which reflects the participants’ inherent skin characteristics. These findings highlight that the WC moisture conditions consistently enhanced the quality of FTIR fingerprint images, regardless of individual skin characteristics among participants. Finally, the analysis of TP minutiae points for the index finger, reflecting the survival rate under different light source conditions, showed no quality differences between lighting conditions in UC and WC.

However, this finding does not conflict with the overall analysis results of this study, as the results for the thumb under the same experimental settings reveal expected outcomes, further supporting the conclusion that light source conditions have a significant impact on FTIR fingerprint image quality. This exceptional result for the index finger is likely due to either the DC moisture condition or inherent differences between the index finger and the thumb. Given that other analyses consistently demonstrate the negative impact of the DC moisture condition on the FTIR fingerprint image quality, this discrepancy is more likely attributed to differences between the thumb and index finger. Specifically, the relatively smaller fingerprint pattern area of the index finger compared to the thumb, along with the absolute number of minutiae points, likely caused the changes in the number of TP points under experimental conditions to be less significant for the index finger.

Another limitation of this study is the lack of precise control over the fingertip contact angle or pressure applied by participants during fingerprint acquisition. Specifically, pressure plays a crucial role in FTIR fingerprint imaging, as it directly affects the contact area between the fingertip and quartz plate, influencing ridge visibility and overall image quality [3]. Although participants were instructed to apply uniform and moderate pressure on the quartz plate during the experiment, individual variations in pressure could have influenced the fingerprint image quality. Future studies should consider implementing a controlled mechanism to regulate finger pressure and contact angle for more consistent image acquisition.

Even though this study included 20 participants, providing valuable insights into the impact of different moisture and light source conditions on FTIR fingerprint image quality, we acknowledge that the sample size is relatively small. According to the central limit theorem (CLT), a sufficiently large sample size is necessary to approximate a normal distribution and ensure the reliability of statistical inferences. Given that our sample size may not fully meet this criterion, the generalizability of our findings is limited [35]. Therefore, caution should be exercised when interpreting the results of this study, as the statistical validity and robustness of the findings may be affected by the limited sample size. A larger sample size in future research would help enhance the robustness and statistical power of the results. Expanding the dataset to include a more diverse population could also improve the validity of the findings.

## 5. Conclusions

In this study, the quality differences in FTIR fingerprint images under varying moisture conditions (UC, DC, and WC) and lighting conditions (R, G, and B) were quantitatively evaluated using an in-house FTIR fingerprint imaging device. Statistical analysis was conducted employing two methods: the number of TP fingerprint minutiae points with a quality index of 0.9 or higher and the number of TP feature points reflecting survival rates. The analysis provided statistical evidence that WC moisture conditions consistently enhance the quality of FTIR fingerprint images, irrespective of the fingerprint donor’s personal skin characteristics. Conversely, it was observed that FTIR fingerprint image quality under DC moisture conditions may be similar to or worse than that of the UC control group, which received no treatment. For light source conditions, the analysis confirmed that the G and B light sources consistently improved FTIR fingerprint image quality compared to the R light source.

These findings hold the potential to contribute to the development of a consistent and standardized measurement method that reduces the influence of individual variability in fingerprint providers. Specifically, adopting an excessive moisture condition (WC) for fingerprint acquisition can help mitigate inconsistencies arising from differences in individual skin properties. Additionally, using G or B illumination can further enhance fingerprint image quality by optimizing contrast and minimizing subsurface scattering. Establishing such standardized imaging conditions is expected to improve the accuracy and reliability of FTIR-based AFIS performance evaluation.

However, this study has some limitations. First, while we confirmed that the B and G light sources are more effective than the R light source, we were unable to precisely determine the qualitative differences between B and G light in terms of their impact on FTIR fingerprint image quality. Second, this study did not account for variations in the fingertip contact angle and pressure applied by participants during fingerprint acquisition, which may have influenced the contact area between the fingertip and the quartz plate, ultimately affecting image quality. Lastly, the relatively small number of participants limits the statistical validity and generalizability of our findings.

To address these limitations, future studies should refine the segmentation of B and G light wavelengths at a nanometer scale to better differentiate their effects on fingerprint image quality. Additionally, illumination intensity should be considered as a variable to further understand its influence. Moreover, incorporating a fingertip placement guide and pressure sensors into the quartz plate support system could help regulate and standardize the force applied to the FTIR fingerprint sensor, thereby reducing variability in image acquisition. Finally, conducting experiments with a larger sample size will enhance the statistical reliability and robustness of future findings.

## Figures and Tables

**Figure 1 sensors-25-01276-f001:**
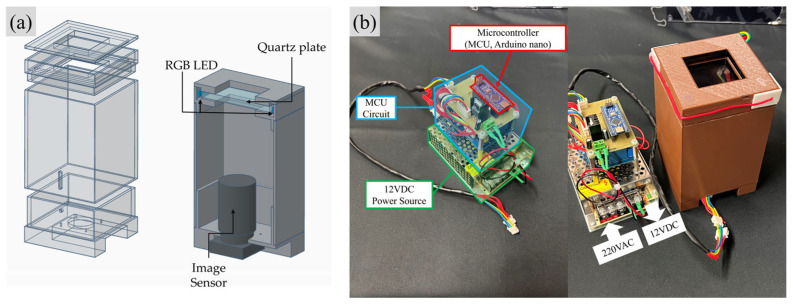
In-house FTIR fingerprint imaging device. (**a**) 3D-print structure blueprint. (**b**) 3D-printed in-house FTIR fingerprint imaging device.

**Figure 2 sensors-25-01276-f002:**
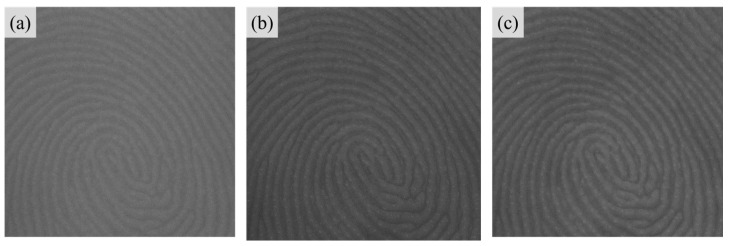
Extracted valid channel data from the image corresponding to its lighting condition. (**a**) R channel data. (**b**) G channel data. (**c**) B channel data.

**Figure 3 sensors-25-01276-f003:**
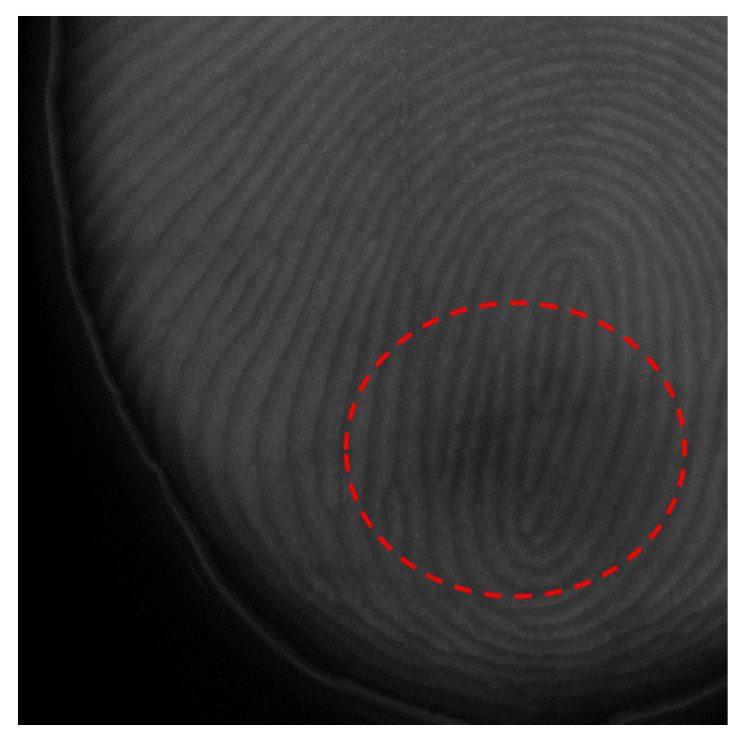
Local brightness error in the FTIR fingerprint images (highlighted areas marked with red circles).

**Figure 4 sensors-25-01276-f004:**
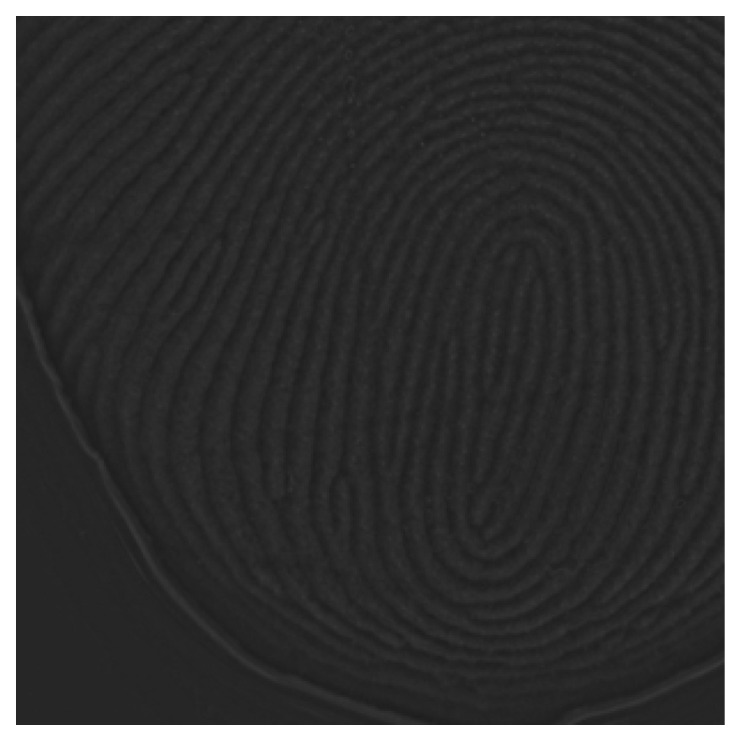
FTIR fingerprint image with local brightness error corrected.

**Figure 5 sensors-25-01276-f005:**
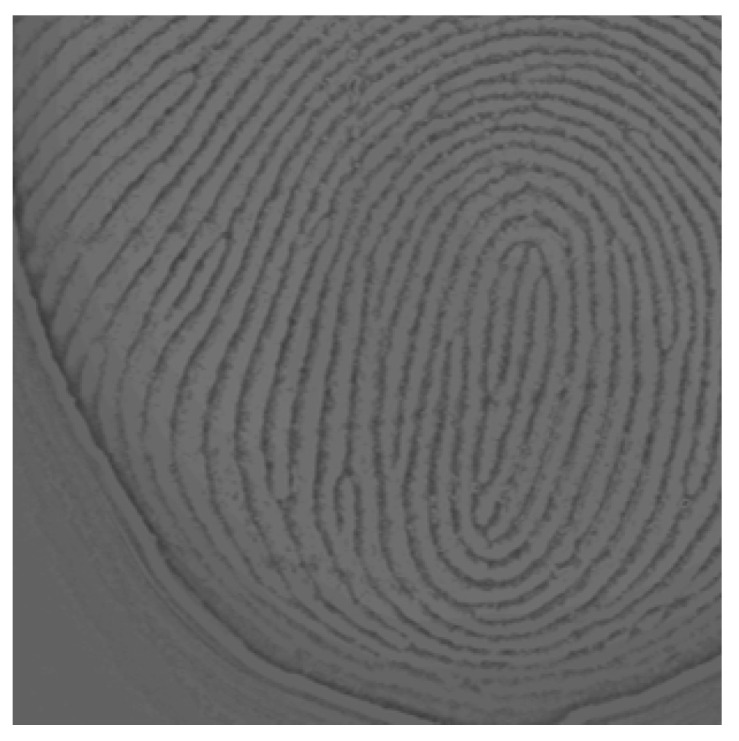
Normalized FTIR fingerprint image.

**Figure 6 sensors-25-01276-f006:**
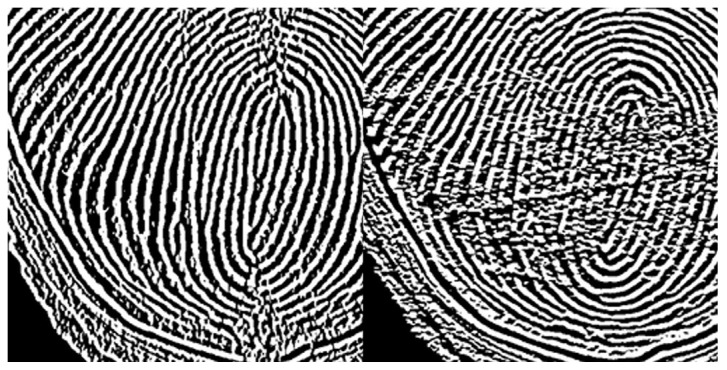
Gradient images of FTIR fingerprint image: x-axis direction gradient (**left**) and y-axis direction gradient (**right**).

**Figure 7 sensors-25-01276-f007:**
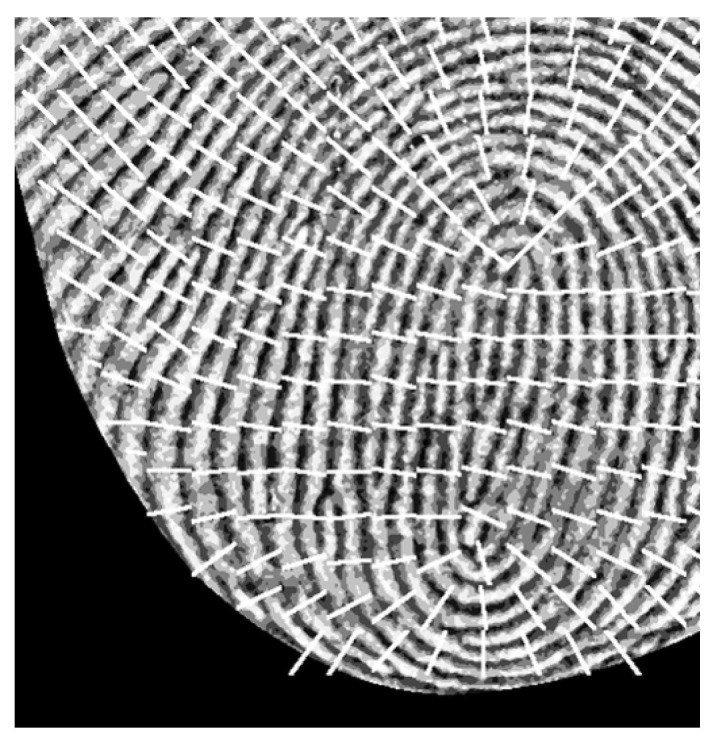
Visualized ridge orientation map.

**Figure 8 sensors-25-01276-f008:**
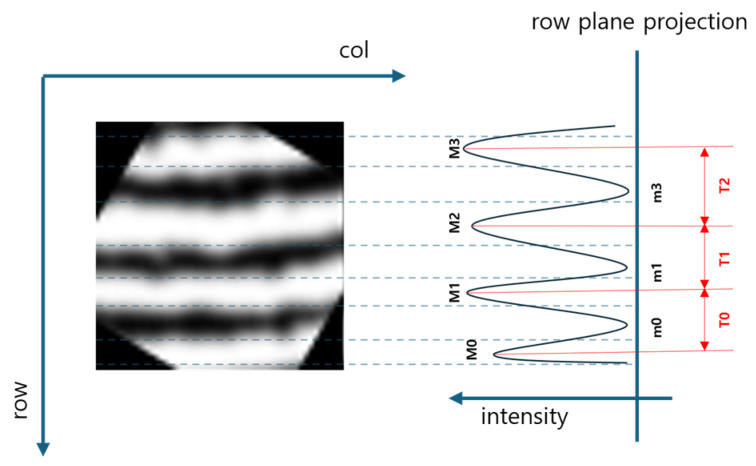
Projection of the segmented FTIR fingerprint image.

**Figure 9 sensors-25-01276-f009:**
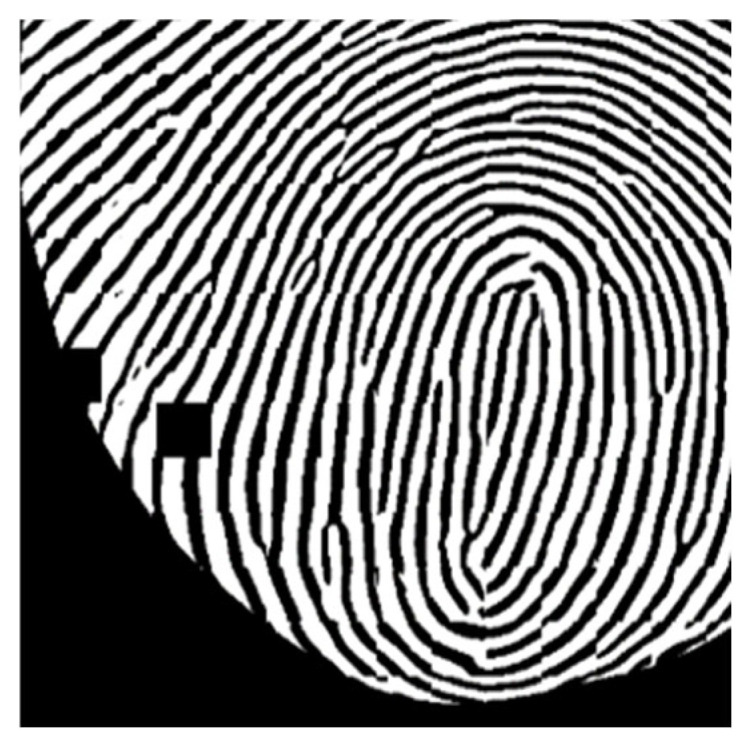
Enhanced fingerprint ridge pattern by Gabor kernel.

**Figure 10 sensors-25-01276-f010:**
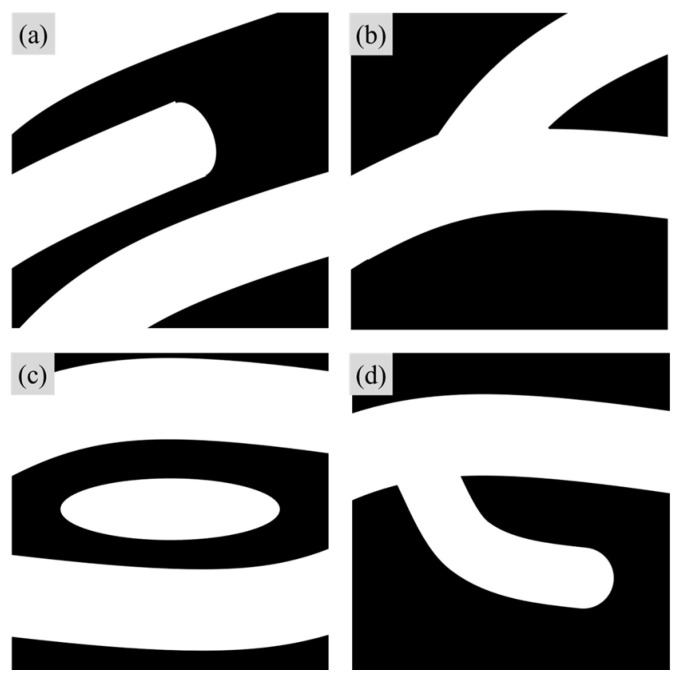
Generic types of fingerprint minutiae. (**a**) ridge ending. (**b**) ridge bifurcation. (**c**) ridge island. (**d**) ridge spur.

**Figure 11 sensors-25-01276-f011:**
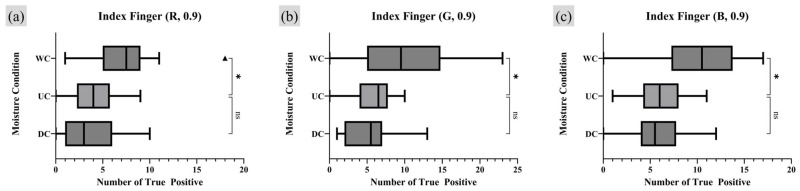
Moisture condition analysis on index finger, RM one-way ANOVA test result (number of TP minutiae points of quality index 0.9 or higher). Markers (triangle, circle, square) represent outlier data. Statistical significance is denoted as follows: ns (not significant) *p* > 0.05; * *p* ≤ 0.05; ** *p* ≤ 0.01; *** *p* ≤ 0.001; **** *p* ≤ 0.0001.

**Figure 12 sensors-25-01276-f012:**
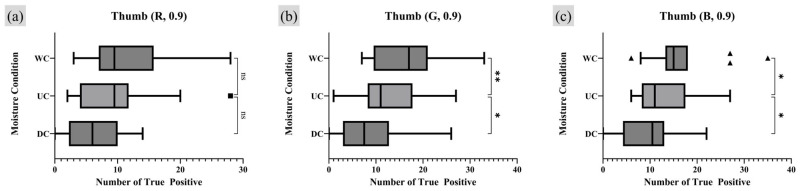
Moisture condition analysis on thumb, RM one-way ANOVA test result (number of TP minutiae points of quality index 0.9 or higher). Markers (triangle, circle, square) represent outlier data. Statistical significance is denoted as follows: ns (not significant) *p* > 0.05; * *p* ≤ 0.05; ** *p* ≤ 0.01; *** *p* ≤ 0.001; **** *p* ≤ 0.0001.

**Figure 13 sensors-25-01276-f013:**
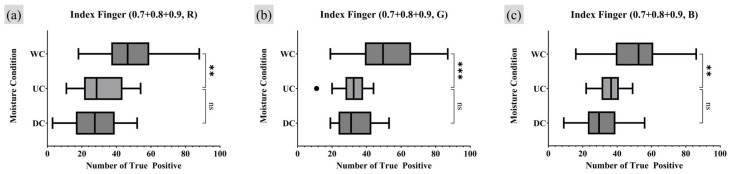
Moisture condition analysis on index fingers RM one-way ANOVA test result (survival rate of TP minutiae point quality index of 0.7 to 0.9 or higher). Markers (triangle, circle, square) represent outlier data. Statistical significance is denoted as follows: ns (not significant) *p* > 0.05; * *p* ≤ 0.05; ** *p* ≤ 0.01; *** *p* ≤ 0.001; **** *p* ≤ 0.0001.

**Figure 14 sensors-25-01276-f014:**
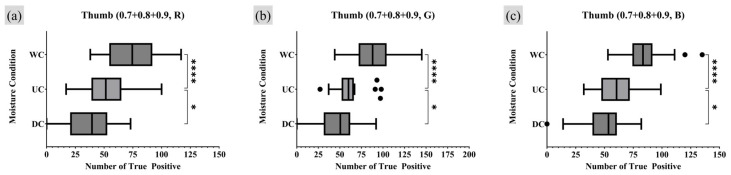
Moisture condition analysis on thumb, RM one-way ANOVA test result (survival rate of TP minutiae point quality index 0.7 to 0.9 or higher). Markers (triangle, circle, square) represent outlier data. Statistical significance is denoted as follows: ns (not significant) *p* > 0.05; * *p* ≤ 0.05; ** *p* ≤ 0.01; *** *p* ≤ 0.001; **** *p* ≤ 0.0001.

**Figure 15 sensors-25-01276-f015:**
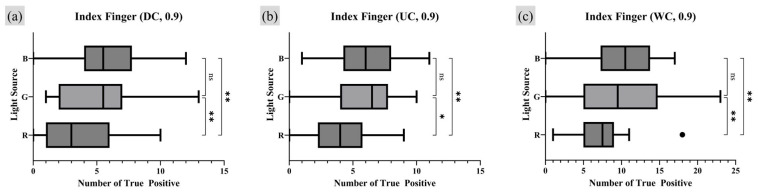
Light source condition analysis on index finger RM one-way ANOVA test result (number of TP minutiae points of quality index 0.9 or higher). Markers (triangle, circle, square) represent outlier data. Statistical significance is denoted as follows: ns (not significant) *p* > 0.05; * *p* ≤ 0.05; ** *p* ≤ 0.01; *** *p* ≤ 0.001; **** *p* ≤ 0.0001.

**Figure 16 sensors-25-01276-f016:**
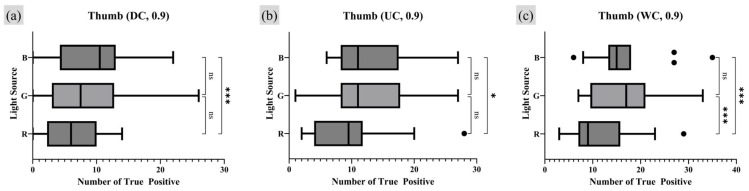
Light source condition analysis on thumb, RM one-way ANOVA test result (number of TP minutiae points of quality index 0.9 or higher). Markers (triangle, circle, square) represent outlier data. Statistical significance is denoted as follows: ns (not significant) *p* > 0.05; * *p* ≤ 0.05; ** *p* ≤ 0.01; *** *p* ≤ 0.001; **** *p* ≤ 0.0001.

**Figure 17 sensors-25-01276-f017:**
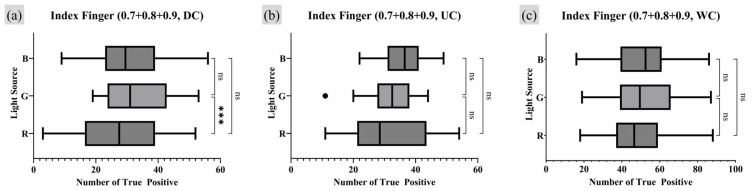
Light source condition analysis on index finger, RM one-way ANOVA test result (survival rate of TP minutiae point quality index from 0.7 to 0.9 or higher). Markers (triangle, circle, square) represent outlier data. Statistical significance is denoted as follows: ns (not significant) *p* > 0.05; * *p* ≤ 0.05; ** *p* ≤ 0.01; *** *p* ≤ 0.001; **** *p* ≤ 0.0001.

**Figure 18 sensors-25-01276-f018:**
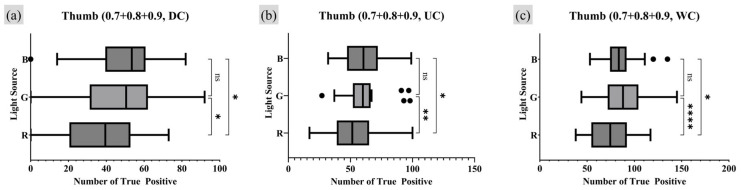
Light source condition analysis on thumb, RM one-way ANOVA test result (survival rate of TP minutiae point quality index from 0.7 to 0.9 or higher). Markers (triangle, circle, square) represent outlier data. Statistical significance is denoted as follows: ns (not significant) *p* > 0.05; * *p* ≤ 0.05; ** *p* ≤ 0.01; *** *p* ≤ 0.001; **** *p* ≤ 0.0001.

**Table 1 sensors-25-01276-t001:** Dunnett’s multiple comparison on index finger (number of TP minutiae points of quality index 0.9 or higher).

Light Source Condition	Control vs. Test	Mean (Control)	Mean (Test)	Mean Diff. ^2^ (Control–Test)	SE ^1^ of Diff.
**R**	UC vs. DC	4.30	3.45	0.85 (ns)	0.89
	UC vs. WC		7.20	−2.90 (*)	1.03
**G**	UC vs. DC	5.65	5.45	0.20 (ns)	0.84
	UC vs. WC		9.50	−3.85 (*)	1.40
**B**	UC vs. DC	6.30	5.60	0.70 (ns)	0.92
	UC vs. WC		9.70	−3.40 (*)	1.21

^1^ Standard Error. ^2^ ns (not significant) *p* > 0.05; * *p* ≤ 0.05; ** *p* ≤ 0.01; *** *p* ≤ 0.001; **** *p* ≤ 0.0001.

**Table 2 sensors-25-01276-t002:** Dunnett’s multiple comparison on thumb (number of TP minutiae points of quality index 0.9 or higher).

Light Source Condition	Control vs. Test	Mean (Control)	Mean (Test)	Mean Diff. ^2^ (Control–Test)	SE ^1^ of Diff.
**R**	UC vs. DC	9.80	6.30	3.50 (ns)	1.53
	UC vs. WC		11.95	−2.15 (ns)	1.92
**G**	UC vs. DC	12.45	8.35	4.10 (*)	1.68
	UC vs. WC		17.30	−4.85 (**)	1.54
**B**	UC vs. DC	13.35	9.60	3.75 (*)	1.29
	UC vs. WC		16.55	−3.20 (*)	1.21

^1^ Standard Error. ^2^ ns (not significant) *p* > 0.05; * *p* ≤ 0.05; ** *p* ≤ 0.01; *** *p* ≤ 0.001; **** *p* ≤ 0.0001.

**Table 3 sensors-25-01276-t003:** Dunnett’s multiple comparison on index finger (survival rate of TP minutiae point quality index of 0.7 to 0.9 or higher).

Light Source Condition	Control vs. Test	Mean (Control)	Mean (Test)	Mean Diff. ^2^ (Control–Test)	SE ^1^ of Diff.
**R**	UC vs. DC	30.90	26.70	4.20 (ns)	3.58
	UC vs. WC		48.30	−17.40 (**)	4.60
**G**	UC vs. DC	32.20	33.35	−1.15 (ns)	2.54
	UC vs. WC		51.70	−19.50 (***)	4.65
**B**	UC vs. DC	32.25	31.45	3.80 (ns)	2.30
	UC vs. WC		51.70	−16.45 (**)	4.34

^1^ Standard Error. ^2^ ns (not significant) *p* > 0.05; * *p* ≤ 0.05; ** *p* ≤ 0.01; *** *p* ≤ 0.001; **** *p* ≤ 0.0001.

**Table 4 sensors-25-01276-t004:** Dunnett’s multiple comparison on thumb (survival rate of TP minutiae point quality index 0.7 to 0.9 or higher).

Light Source Condition	Control vs. Test	Mean (Control)	Mean (Test)	Mean Diff. ^2^ (Control–Test)	SE ^1^ of Diff.
**R**	UC vs. DC	52.50	39.50	13.00 (*)	5.44
	UC vs. WC		75.75	−23.25 (****)	4.36
**G**	UC vs. DC	62.15	47.95	14.20 (*)	4.58
	UC vs. WC		88.45	−26.30 (****)	4.13
**B**	UC vs. DC	61.65	48.65	13.00 (*)	4.43
	UC vs. WC		85.00	−23.35 (****)	3.25

^1^ Standard Error. ^2^ ns (not significant) *p* > 0.05; * *p* ≤ 0.05; ** *p* ≤ 0.01; *** *p* ≤ 0.001; **** *p* ≤ 0.0001.

**Table 5 sensors-25-01276-t005:** Tukey’s multiple comparison on index finger (number of TP minutiae points of quality index 0.9 or higher).

Moisture Condition	Group 1 vs. Group 2	Mean (Group 1)	Mean (Group 2)	Mean Diff. ^2^ (Group 1–Group 2)	SE ^1^ of Diff.
**DC**	R vs. G	3.45	5.45	−2.00 (**)	0.57
	R vs. B	3.45	5.60	−2.15 (**)	0.61
	G vs. B	5.45	5.60	−0.15 (ns)	0.65
**UC**	R vs. G	4.30	5.65	−1.35 (*)	0.51
	R vs. B	4.30	6.30	−2.00 (**)	0.56
	G vs. B	5.65	6.30	−0.65 (ns)	0.40
**WC**	R vs. G	7.20	9.50	−2.30 (**)	0.68
	R vs. B	7.20	9.70	−2.50 (**)	0.59
	G vs. B	9.50	9.70	−0.20 (ns)	0.61

^1^ Standard Error. ^2^ ns (not significant) *p* > 0.05; * *p* ≤ 0.05; ** *p* ≤ 0.01; *** *p* ≤ 0.001; **** *p* ≤ 0.0001.

**Table 6 sensors-25-01276-t006:** Tukey’s multiple comparison on thumb (number of TP minutiae points of quality index 0.9 or higher).

Moisture Condition	Group 1 vs. Group 2	Mean (Group 1)	Mean (Group 2)	Mean Diff. ^2^ (Group 1–Group 2)	SE ^1^ of Diff.
**DC**	R vs. G	6.30	8.35	−2.05 (ns)	0.88
	R vs. B	6.30	9.60	−3.30 (***)	0.71
	G vs. B	8.35	9.60	−1.25 (ns)	0.61
**UC**	R vs. G	9.80	12.45	−2.54 (ns)	1.08
	R vs. B	9.80	13.35	−3.55 (*)	1.13
	G vs. B	12.45	13.35	−0.90 (ns)	0.95
**WC**	R vs. G	11.95	17.30	−5.35 (***)	1.17
	R vs. B	11.95	16.55	−4.60 (***)	0.96
	G vs. B	17.30	16.55	−0.75 (ns)	0.91

^1^ Standard Error. ^2^ ns (not significant) *p* > 0.05; * *p* ≤ 0.05; ** *p* ≤ 0.01; *** *p* ≤ 0.001; **** *p* ≤ 0.0001.

**Table 7 sensors-25-01276-t007:** Tukey’s multiple comparisons on index fingers (survival rate of TP minutiae point quality index from 0.7 to 0.9 or higher).

Moisture Condition	Group 1 vs. Group 2	Mean (Group 1)	Mean (Group 2)	Mean Diff. ^2^ (Group 1–Group 2)	SE ^1^ of Diff.
**DC**	R vs. G	26.70	33.35	−6.65 (***)	1.35
	R vs. B	26.70	31.45	−4.75 (ns)	2.14
	G vs. B	33.35	31.45	1.90 (ns)	1.85
**UC**	R vs. G	30.90	32.20	−1.30 (ns)	2.13
	R vs. B	30.90	35.25	−4.35 (ns)	2.10
	G vs. B	32.20	35.25	−3.05 (ns)	1.58
**WC**	R vs. G	48.30	51.70	−3.40 (ns)	2.46
	R vs. B	48.30	51.70	−3.40 (ns)	2.48
	G vs. B	51.70	51.70	−0.00 (ns)	2.01

^1^ Standard Error. ^2^ ns (not significant) *p* > 0.05; * *p* ≤ 0.05; ** *p* ≤ 0.01; *** *p* ≤ 0.001; **** *p* ≤ 0.0001.

**Table 8 sensors-25-01276-t008:** Tukey’s multiple comparison on thumb (survival rate of TP minutiae point quality index from 0.7 to 0.9 or higher).

Moisture Condition	Group 1 vs. Group 2	Mean (Group 1)	Mean (Group 2)	Mean Diff. ^2^ (Group 1–Group 2)	SE ^1^ of Diff.
**DC**	R vs. G	39.50	47.95	−8.45 (*)	2.67
	R vs. B	39.50	48.65	−9.15 (*)	2.84
	G vs. B	47.95	48.65	−0.7 (ns)	2.02
**UC**	R vs. G	52.50	62.15	−9.65 (**)	2.54
	R vs. B	52.50	61.65	9.15 (*)	2.98
	G vs. B	62.15	61.65	0.50 (ns)	2.63
**WC**	R vs. G	75.75	88.45	−12.70 (****)	2.13
	R vs. B	75.75	85.00	−9.25 (*)	2.96
	G vs. B	88.45	85.00	3.45 (ns)	2.92

^1^ Standard Error. ^2^ ns (not significant) *p* > 0.05; * *p* ≤ 0.05; ** *p* ≤ 0.01; *** *p* ≤ 0.001; **** *p* ≤ 0.0001.

## Data Availability

The data used in this study are partially available in the form of anonymized processed data to protect participants’ sensitive personal information under the supervision of the Institutional Review Board of Dongguk University. All raw data used in this study will be stored securely until 14 June 2027, as directed by the Dongguk University Institutional Review Board, and then completely deleted.

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
