# Peer review of "A Quantitative Analysis Study on the Effects of Moisture and Light Source on FTIR Fingerprint Image Quality"

_sensors, 2025, doi:10.3390/s25041276_

Round 1

Reviewer 1 Report

Comments and Suggestions for Authors

1.Although the study used 20 participants, the sample size is still relatively small and may not fully represent a wider population. Suggest increasing the sample size in future research to improve the generalizability and reliability of the results.

2.The article found that blue-green light sources (B, G) are more effective in improving fingerprint image quality than red light sources (R), but did not delve into the underlying physical mechanisms. Suggest adding a theoretical analysis of the relationship between light source wavelength and fingerprint image quality in the discussion section to enhance the depth of the article.

3.The article mentions that due to the protection of participant privacy, the original data cannot be made public. Although this is reasonable, it is recommended to provide partially anonymized data or code so that other researchers can reproduce the experimental results.

Comments on the Quality of English Language

1.The structure of the abstract section is relatively chaotic and the logic is not coherent enough.

Author Response

Please find the attached file for your review.

Reviewer 2 Report

Comments and Suggestions for Authors

The reviewed work focuses on study of the dependence of the quality of fingerprint images obtained by the Frustrated Total Internal Reflection (FTIR) method on the humidity of the surface and the wavelength of the radiation. My comments thereupon are provided below:

1. This work considers methods for improving fingerprint images obtained by the FTIR method that optimise the parameters of the registration system, but the state of the object also affects image quality, and this needs to be commented on. It is necessary to control (or fix) the degree of finger pressure on the surface, control the absence of any films on the finger surface (or on the imaging surface). To improve fingerprint images, it is not only necessary to improve the photography conditions but also to improve the preparation of the object for photography. This needs to be discussed.

2. The main results of this work consist in finding the best conditions for humidity and the best colour for photography. But what about the illumination intensity? What is the optimum here? The same applies to the spectral width of the illuminating radiation. What would be the optimal spectral width? This needs to be discussed.

3. The studies were conducted with 20 participants, but this is a small sample for general conclusions. Accordingly, the results of the work can be evaluated as preliminary with insufficient statistical proof. In this regard, it is necessary to correct the words about “strong statistical evidence” in the abstract.

4. What are the conclusions from the reported results? Should the number of participants be greater to obtain more statistically substantiated data? Should the optimal humidity of the air be ensured in the place where fingerprint images are obtained? Should the optimal colour of the backlight be ensured when obtaining fingerprint images? It is necessary to formulate recommendations based on the results of the research conducted in this work.

If the Authors take into account the above comments when revising the article, it may be published in Sensors.

Round 2

Reviewer 2 Report

Comments and Suggestions for Authors

In response to my observations, important information was added to the manuscript that made it

more interesting and comprehensible. My comments have been fully addressed by the Authors in

the revised manuscript, which may be now published.